# Diagnostic, prognostic, and therapeutic potentials of gut microbiome profiling in human schistosomiasis: A comprehensive systematic review

Martin Gael Oyono[1,2,3], Sebastien Kenmoe[4], Jean Thierry Ebogo Belobo[1], Leonel Javeres Mbah Ntepe[2], Mireille Kameni[2,5], Leonel Meyo Kamguia[2], Thabo Mpotje[6], Justin Komguep Nono [2,7] *

1 Laboratory of Microbiology, Infectious Diseases and Immunology, Institute of Medical Research and Medicinal Plant Studies (IMPM), Ministry of Scientific Research and Innovation, Yaoundé, Cameroon, 2 Unit of Immunobiology and helminth infections, Laboratory of Molecular Biology and Biotechnology, Institute of Medical Research and Medicinal plant Studies, Ministry of Scientific Research and Innovation, Yaoundé, Cameroon, 3 Laboratory of Parasitology and Ecology, Department of Animal Biology and Physiology, University of Yaoundé 1, Yaoundé, Cameroon, 4 Department of Microbiology and Parasitology, University of Buea, Buea, Cameroon, 5 Department of Microbiology and Parasitology, University of Bamenda, Bamenda, Cameroon, 6 Africa Health Research Institute, Durban, Kwazulu-Natal, South Africa, 7 Division of Immunology, Faculty of Health Sciences, University of Cape Town, Cape Town, South Africa

* justkoms@yahoo.fr

## Abstract

### Background

Several studies have highlighted alteration in the gut microbiome associated with the onset and progression of diseases. Recognizing the potential of gut microbiota as biomarkers, this systematic review seeks to synthesize current data on the intricate relationship between the host gut microbiome profiles and their usefulness for the development of diagnostic, prognostic and therapeutic approaches to control human schistosomiasis.

### Methods

A systematic literature review was carried out by searching for relevant studies published until date, that is May 2024, using Medline, Embase, Global Health, Web of Science, and Global Index Medicus databases. The keywords used to select articles were "Gut microbiome", "Gut Microbiota", "Schistosomiasis", "Bilharziasis ", and "Human". Extracted data were analysed qualitatively from the selected articles.

### Results

Of the 885 articles retrieved and screened, only 13 (1.47%) met the inclusion criteria and were included in this review. Of the included studies, 6 (46.2%) explored alterations of gut microbiome in schistosome-infected patients, 4 (30.7%) in patients with liver pathologies, and 3 (23.1%) in patients treated with praziquantel. Bacteria from the genera *Bacteroides*, *Faecalibacterium*, *Blautia* and *Megasphaera* were associated with *S. japonicum* and *S.*

**Data Availability Statement:** All data in the form of online references of the exploited articles are fully

available within the manuscript and supporting documents.

**Funding:** The author(s) received no specific funding for this work.

**Competing interests:** JKN is a recipient of a Merck KGaA Global Health Institute research grant. MGO, MK, LJMN and LMK are supported by fellowships through this Merck KGaA research fund to JKN. JKN is a founding member of JRJ Health, a health-promoting association based in Cameroon. JRJ had no role in the conceptualization, design, analysis of collected data, decision to publish, and preparation of the manuscript. All the remaining authors declare no financial or non-financial competing interests.

*haematobium* infection in school-aged children, whereas infection with *S. mansoni* rather associated with *Klebsiella* and *Enterobacter*. The gut microbiota signature in patient with schistosomiasis-induced liver pathology was reported only for *S. japonicum*, and the genus *Prevotella* appeared as a non-invasive biomarker of *S. japonicum*-associated liver fibrosis. For *S. mansoni*-infected school-aged children, it further appeared that the treatment outcome following praziquantel administration associated with the abundance in the gut microbiome of bacteria from the classes Fusobacteriales, Rickettsiales and Neisseriales.

## Conclusion

The host gut microbiome appears to be a valuable, non-invasive, but still poorly utilized, source of host biomarkers potentially informative for better diagnosing, prognosing and treating schistosomiasis. Further studies are therefore needed to comprehensively define such gut microbial biomarkers of human schistosomiasis and catalyse the informed development of gut microbiome-based tools of schistosomiasis control.

### Author summary

Schistosomiasis orchestrates profound regulation by/of the host immune responses. Given the core role of the gut microbiome in regulating the human immune responses at the systemic level, and the increasingly recognized potential of this host component in biomarking for the identification, monitoring, prevention and treatment of several diseases, we questioned whether the human gut microbiome shows patterns of alterations during schistosomiasis that could robustly inform novel diagnostic, prognostic and therapeutic approaches.

After systematically scrutinizing all available reports in the literature on the gut microbiome dynamics during schistosomiasis in humans so far, available evidences pointed at pathognomonic changes in human gut microbiomes following schistosomiasis infection, progression as pathology develops and treatment. Despite the fact that evidences from most of these studies are not yet definitive i.e. hardly controlled for poly-infections, for diet-driven alterations or for robustness of the observed signatures in large longitudinal studies, some of the gut microbial changes observed so far presented significantly and repeatedly in several individuals and at times in independent studies. This indicates here an exploitable potential for the gut microbiome that warrants more comprehensive and controlled studies to unequivocally identify diagnostic, prognostic and therapeutic gut microbial biomarkers during human schistosomiasis.

## Introduction

Schistosomiasis, also known as bilharzia or Katayama fever, is a parasitic disease caused by trematode helminths of the genus *Schistosoma*, which live and reproduce sexually in the human circulatory system [1]. This neglected tropical disease (NTD) affects at least 251.4 million in 78 countries worldwide. Approximately 800 million people were at risk, with Sub-Saharan Africa (SSA) bearing more than 90% of the global burden of the disease [2]. This is reflected in the severe impact of schistosomiasis, which leads to an estimated loss of 1.6 million Disability-Adjusted Life Years (DALYs) and causes 150–280,000 deaths annually in the region

[3]. The highest prevalence and intensity of the disease is observed among school-aged children [4].

*S. haematobium*, *S. mansoni* and *S. japonicum* are the three main species of *schistosomes* that infect humans. *S. haematobium* and *S. mansoni* are found in Africa and the Middle East, while only *S. mansoni* occurs in America. *S. japonicum* is found mainly in Asia, especially in the Philippines and China. In addition, human schistosomiasis can also be due to three locally endemic species: *S. mekongi* in the Mekong basin, *S. guineensis* and *S. intercalatum* in West and Central Africa. The distribution of each species is determined by its host snail range, as each species has a specific range of suitable snail hosts [4].

Infection occurs when cercariae, the larval stage of the parasite shed by freshwater snails, penetrates the skin, bypasses the immune system, develop into schistosomula, gain access to the blood stream and reach the hepatic portal vein. Following sexual maturation, the female and male schistosomes pair migrates against the blood stream and reach their final destination: the mesenteric vein of the intestine (*S. mansoni* and *S. japonicum*) or the pelvic venous plexus (*S. haematobium*), the site of eggs deposition [1]. The intravascular eggs traverse the wall of the venule, pass through the surrounding tissue and reach the lumen of the intestine (*S. mansoni* and *S. japonicum*) or bladder (*S. haematobium*), where they are expelled in faeces or urine, respectively [1,4–5]. Some eggs become trapped in host tissues, causing the main pathology during schistosomiasis (i.e., liver fibrosis or genital schistosomiasis).

The immune system of infected hosts is challenged by multiple schistosome life cycle stages such as penetrating cercariae, adult worms and eggs. As a result, a multitude of humoral and cellular immune responses are stimulated leading the antibody production and T-cell proliferation targeting specific parasite stages. There are generally three stages of schistosomiasis, beginning with an initial dermatitis reaction following contamination, resulting in an allergic, inflammatory maculopapular lesion [6]. The acute symptomatic phase characterized by a T helper 1 cells (Th1) response follows this initial phase, whose most common symptoms are persistent fever (Katayama fever), weakness, vomiting, nausea, diarrhoea, malaise and rapid weight loss [1]. With the progression of the diseases, when the first eggs are produced by recently matured worms, the host immune response strongly become T helper 2 cells (Th2)-polarized. This shift results in the loss of pre-existing worm antigen specific Th1 response, leading to the onset of chronic schistosomiasis, the final phase of the disease [7]. During this immune transition, the toxic effects of trapped schistosome eggs' secretome are neutralized through the formation of granulomas that encapsulate parasite eggs to prevent tissue damage. This process is driven by an immune-mediated CD4+ T-cell-dependent response that leads to granuloma formation [8]. However, this immune response triggers eosinophilic inflammatory reactions, which are gradually replaced by fibrotic deposition and may lead to cancer [9]. In hepatic schistosomiasis this host's cell-mediated immune response might result in irreversible fibrosis and severe portal hypertension, if left untreated [10].

The current strategy for controlling schistosomiasis is based on an annual campaign of mass administration of praziquantel (PZQ), as no effective vaccine has yet been developed [11]. In 2020, the WHO released its roadmap for addressing NTDs globally and sets ambitious targets to eliminate schistosomiasis as a public health problem by 2030 [12]. To achieve elimination goals, it is crucial to develop simple, reliable, and minimal invasive tools for accurately detecting *Schistosoma* eggs in stools or urine samples, and associated pathologies. Research into identifying and validating appropriate biomarkers, such as signature from the gut microbiome, to indicate the presence and progression of *Schistosoma* infection is strongly encouraged [13,14]. The human gastrointestinal (GI) tract harbours a complex and dynamic community of microorganisms known as the gut microbiota, which significantly impacts the host's health both during homeostasis and disease [15]. Multiple factors (e.g., diet) play a role

in shaping the human gut microbiota across the lifetime. The gut microbiota provides numerous benefits to the host, including enhancing gut integrity, shaping the intestinal epithelium through epithelial cell proliferation and differentiation, harvesting energy, protecting form pathogens, and regulating host immunity [16,17]. However, dysbiosis, or alteration in the composition and function of the microbial composition, has been linked to a variety of GI and systemic disorders, including obesity, colorectal cancer, cardiovascular risk, autism and Alzheimer's disease [18,19]. In developing countries, helminth infections are a common cause of gut microbiota dysbiosis. Several studies reported alterations of gut microbial richness, diversity and modifications in metabolic signatures associated with human gastrointestinal helminth infections [20–23]. Interestingly, only thirteen studies, as retrieved in the present systematic review, have robustly shown that gut microbiota becomes altered during acute and advanced human schistosomiasis. These studies, though limited in their numbers, suggest that gut microbiota may be helpful for the diagnosis, monitoring, treatment, and prevention of schistosomiasis. This review seeks to comprehensively evaluate evidences from these studies and summarize existing findings on the relationship between the gut microbiome and schistosomiasis in humans.

## Methods

This systematic review was designed and conducted using the Preferred Reporting Items for Systematic Reviews and Meta-Analyses (PRISMA) checklist (**S1 Table**) [24]. The protocol was published in the international database PROSPERO under the identification CRD42023425371. Ethical clearance was not required as this systematic review solely reports on previously published data.

### Literature search strategy

A comprehensive search strategy (**S2 Table**) was designed to enable a search of relevant studies in the following databases: Medline, Embase, Global Health, Web of Science, and Global Index Medicus. These databases were searched for studies published up to May 31, 2024 in order to have current information. To find any new potential research, the reference lists of the articles that were uncovered during the literature search were examined, along with other sources such as websites. The search terms used were as follows: ("Gut microbiome" or "Gastrointestinal Microbiome" or "Gut microflora" or "Gut Microbiota" or "Gastrointestinal Flora" or "Gut Flora" or "dysbiosis") AND ("Schistosomiasis" or "Schistosoma Infection" or "Katayama Fever" or "Bilharziasis").

### Study selection

Studies published in English or French languages were included if they contained data about the gut microbiota composition in schistosomiasis patient. After removing duplicated studies from the list, titles and abstracts of eligible findings were independently examined by two authors (MGO and SK) for the selection of relevant studies. In this review, we considered cross-sectional, case-control, and longitudinal studies, as well as community- and hospital-based studies carried out worldwide. We included studies that reported modification of gut microbiota composition and/or function in schistosomiasis-infected patients with or without pathology. Studies with inappropriate study designs, such as comments, case reports, reviews, systematic reviews, and meta-analyses, animal studies, and articles with no data about the gut microbiota composition changes in schistosomiasis patient data were excluded. Studies published in neither the English nor French languages, as well as studies whose complete text and

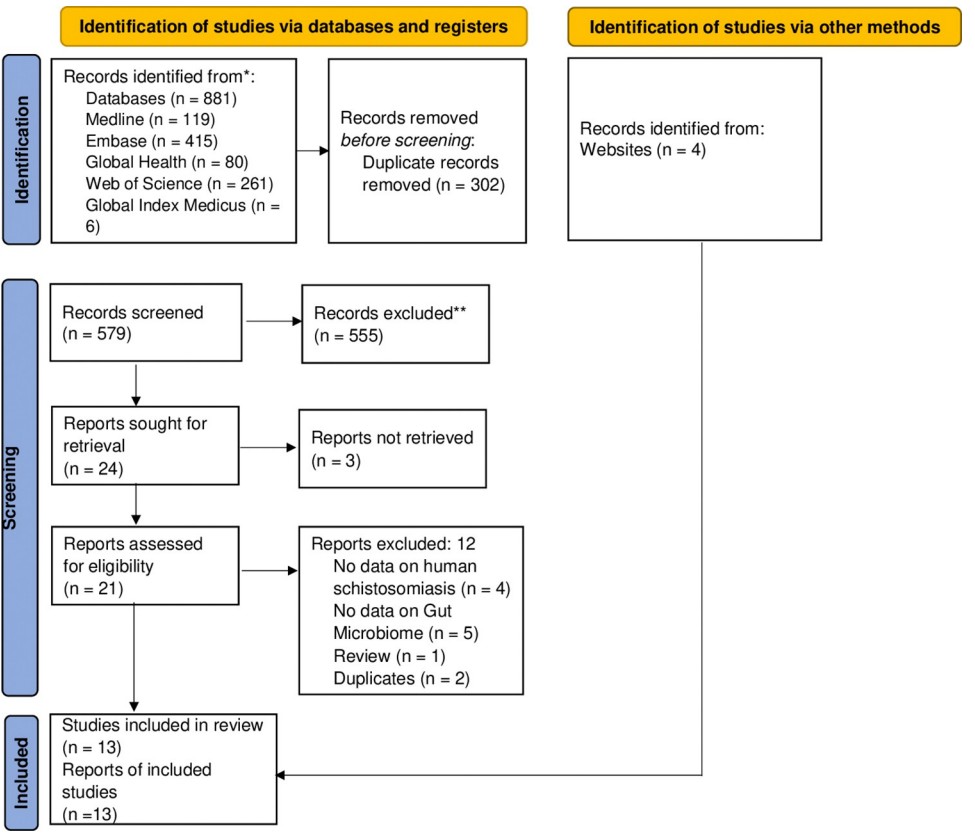

**Fig 1. PRISMA flow diagram showing the study selection process.**

abstract were unavailable or unretrievable, were also excluded. Details of the study selection process are presented in the PRISMA flow diagram (**Fig 1**).

## Data extraction

An online google form was designed by two authors (MGO and SK) and used for data extraction, data from the full text of eligible publications was extracted independently by three authors: MGO, SK, and JTEB. The following variables were extracted for further analysis: first author's name, year of publication, country where the study was carried out, study design, study population, control group, gender, mean age, mean BMI, *Schistosoma* species, infection stage (acute infection, liver fibrosis, liver cirrhosis), schistosomiasis diagnostic tools, microbiota analysis technique (amplicon or shotgun sequencing), diversity modification (alpha and beta), taxa up or down altered, taxa with biomarking potential, study value for schistosomiasis (diagnostic, prognostic or therapeutic), and other biological alterations. Any discrepancies concerning the selection and inclusion of studies and extracted data was resolved by discussion, consensus, or by a fourth author.

## Assessment of risk of bias

The critical appraisal tool developed by Hoy and colleagues was used to assess the quality of the studies [25]. This tool, specifically designed for prevalence studies, evaluates both external and internal validity. If the study was not excluded, a score was calculated for each article and classified as "high", "moderate" or "low" risk of bias (**S3 Table**).

## Results

### Characteristics of the included articles

As shown in Fig 1, the literature search across various databases initially yielded a total of 881 potentially relevant articles. After removing 302 duplicate articles and excluding 555 articles based on thorough screening of the titles and abstracts, 24 articles were sought for retrieval. Of the 24 articles, 3 articles could not be retrieved, leaving 21 articles to be assessed for eligibility. Following the full-text screening, 9 articles met the inclusion criteria. Additionally, 4 relevant articles were identified through manual searches of the reference lists of the included studies, resulting in a total of 13 publications included in this systematic review.

In terms of study design, of the 13 studies included in this review, more than three-quarters (10 studies: 77%) were cross-sectional studies [26–35], the remaining were: 1 (7.7%) longitudinal study [36], 01 (7.7%) clinical trial [37], and 01 (7.7%) case-control [38]. Seven (53.8%) of the studies were hospital-based [28,30,31,33,34,35,38] and 6 (46.2%) were community-based studies [26,27,29,32,36,37]. Most of the study participants were adults (n = 7; 53.8%) [27,28,30,33,34,35,38] followed by children (pre- and school-aged) (n = 6; 46.2%) [27,29,31,32,36,37], and then adolescents (n = 1; 7.7%) [26]. Studies selected in this review were conducted in two schistosomiasis-endemic areas of the world: Asia (7 studies, 53.8%) [27,28,30,33,34,35,38] and SSA (6 studies, 46.2%) studies [26,29,31,32,36,37]. In SSA, where the burden of human schistosomiasis is greatest [2], of the forty SSA countries endemic to human schistosomiasis, studies of the relationship between human schistosomiasis and gut microbiome were conducted only in 3 countries: Zimbabwe (3 studies, 23%) [29,31,32], Nigeria (2 studies, 15.4%) [26,36] and Côte d'Ivoire (1 study, 7.7%) [37]; in Asia, China was the main study site with 6 (46.2%) studies [28,30,33,34,35,38], and the Philippine with 1 (7.7%) study [27].

In addition, the included studies focus on the three main species of human *Schistosoma* including *S. japonicum*, the most reported species (7 studies, 53.8%) [27,28,30,33,34,35,38], followed by *S. haematobium* (5 studies, 38.5%) [26,39,31,32,36] and *S. mansoni* (1 study, 7.7%) [37]. Traditional tools were used for the detection of *Schistosoma* eggs in samples collected: Kato Katz (KK) technique for the detection of *S. japonicum* and *S. mansoni* eggs in the stool sample, and urine filtration for the detection of *S. haematobium* eggs in urine sample. Next generation sequencing methods were used to characterise the gut microbiota signature of the study participants as follows: amplicon (16S rRNA) sequencing (11 studies, 84.6%) [26–30,33–37] and shotgun metagenomics sequencing (2 studies, 15.4%) [31,32]. All the characteristics of the 13 studies included in this systematic review are summarized in **The Table 1** below.

### Alteration in gut microbiome of patients with schistosomiasis infection

Among the 13 articles included in this systematic review, 9 (69.2%) of them reported modifications of gut microbiome structure and composition among patients infected with *S. haematobium* [26,29,31,32,36], *S. japonicum* [27,28,30] and *S. mansoni* [37]. Overall, 4 bacterial genus appeared as potential biomarkers for *S. haematobium* and *S. japonicum* infection namely *Faecalibacterium*, *Bacteroides*, *Blautia* and *Megasphaera* (**Fig 2A**). Details are shown in **Table 2**.

### Gut microbiome signature of *S. haematobium*-infected patients

Alterations of the gut microbiome were observed among *S. haematobium*-infected patients compared to healthy individuals in 4 out of the 13 articles included in the current systematic review. Overall, the gut microbiome of infected patients was dominated by bacteria phyla of *Bacteroidetes*, *Firmicutes*, and *Proteobacteria* [29, 31]. In terms of diversity, there was not a

**Table 1. Characteristics of all included studies.**

| Studies | Study design | Setting | Country | *Schistosoma* species | Description of the study population | Methods of microbiome characterization |
|---|---|---|---|---|---|---|
| [26] | Cross-sectional | Community-based | Nigeria | *Schistosoma haematobium* | 50 adolescents (43 male and 8 female) aged from 11–15 years: 25 infected and 25 control. | Amplicon (16s rRNA) |
| [27] | Cross-sectional | Community-based | Philippines | *Schistosoma japonicum* | 219 participants from 2 cohorts (cohort 1: 161; cohort 2: 58) aged from 4 to 72 years. | Amplicon (16s rRNA) |
| [28] | Cross-sectional | Hospital-based | China | *Schistosoma japonicum* | 26 male fishermen, >30 years old ((15 controls, 11 patients). | Amplicon (16s rRNA) |
| [29] | Cross-sectional | Community-based | Zimbabwe | *Schistosoma haematobium* | 139 children (73 males and 66 females) aged 6 months to 13 years. | Amplicon (16s rRNA) |
| [30] | Cross-sectional | Hospital-based | China | *Schistosoma japonicum* | 26 participants (15 controls, 11 patients) aged more than 30 years old. | Amplicon (16s rRNA) |
| [31] | Cross-sectional | Hospital-based | Zimbabwe | *Schistosoma haematobium* | 113 preschool-aged children (57 males, 56 females), aged 1–5 years with the mean age 3.7 ± 1.1 years. | shotgun metagenomic sequencing |
| [32] | Cross-sectional | Community-based | Zimbabwe | *Schistosoma haematobium* | 116 children (57 males, 59 females) from 1 to 5 years old with the mean age 3.7 ± 1.1 years, 18 infected with *S. haematobium* and 98 uninfected. | shotgun metagenomic sequencing |
| [33] | Cross-sectional | Hospital-based | China | *Schistosoma japonicum* | 29 participants (20 males, 9 females), aged from 46 to 71 years old, 5 with advanced and 24 with chronic *S. japonicum* infection. | Amplicon (16S rRNA) |
| [34] | Cross-sectional | Hospital-based | China | *Schistosoma japonicum* | 41 participants (21 males, 20 females): 20 with chronic infection (53 ± 2.30 years), 8 with advanced infection (56 ± 3.65) and 13 healthy (55 ± 3.24) | Amplicon (16s rRNA) |
| [35] | Cross-sectional | Hospital-based | China | *Schistosoma japonicum* | 40 participants (14 males, 26 females): 10 with liver fibrosis (59.8 ±6.4 years) and 30 healthy (61.1±9.6 years). | Amplicon (16s rRNA) |
| [36] | Longitudinal | Community-based | Nigeria | *Schistosoma haematobium* | 96 children aged between 11–15 years, 46 *S. haematobium* infected (41 males, 5 females) and 50 uninfected. | Amplicon (16s rRNA) |
| [37] | Clinical Trial | Community-based | Ivory Coast | *Schistosoma mansoni* | 34 pre- and school-aged children, from 3 to 13 years old, 28 infected with *S. mansoni* and 6 uninfected. | Amplicon (16S rRNA) |
| [38] | Case control | Hospital-based | China | *Schistosoma japonicum* | 24 patients with liver cirrhosis (12 male and 12 female) with average age 82.7 ± 7.3, and 25 age- and sex-matches healthy (12 male and 13 female) with average age 82.3 ± 7.1 | Amplicon (16s rRNA) |

significant difference in α-diversity of gut microbiota between the schistosomiasis-infected and -uninfected individuals, but the difference was significant when examining β-diversity of gut microbiota. At the level of phylum, the gut microbiome signature of *S. haematobium*-infected patients was characterized by a decrease of the abundance of *Firmicutes*, *Tenericutes*, and *Cyanobacteria*, and a significant increase of the abundance in Proteobacteria, Euryarchaeota [26, 36]. At the lower taxonomy level, mainly at genus level, studies reported a significant increase of *Acinetobacter*, *Treponema*, *Bacteroides*, *Alloprevotella*, *Prevotella*, *Megasphaera*, *Dialister*, *Desulfovibrio*, *Haemophilus*, *Peptococcus*, *Olsenella*, *Pseudomonas*, *Stenotrophomonas*, *Derxia*, *Thalassospira*, and uncultured Coriobacteriaceae; while the key genera that significantly decreased were *Azospirillum*, *Pediococcus*, *Chloroplast*, *Lactobacillus*, *Weissella*, Subdoligranulum, *Parabacteroides*, *Peptostreptococcaceae incertae sedis*, *Clostridium sensu stricto 6*, and uncultured Erysipelotrichaceae uncultured Mollicutes RF9 [26,29,31,36]. See **Table 2**.

Two articles examined alterations in the gut mycobiome during *S. haematobium* infection [31,32]. The results reported that the gut mycobiome made up less than 1% of the sequenced gut microbiome, with a high diversity. The most abundant fungi phyla were Ascomycota (genera: *Protomyces*, *Aspergillus*, *Taphrina*, *Saccharomyces*, *Candida*, *Nakaseomyces*), Microsporidia (genus: *Enterocytozoon*), and Zoopagomycota (genus: *Entomophthora*). Osakunor *et al.* [31] found the abundance of *Aspergillus*, *Tricholoma*, and *Periglandula* higher in infected preschool-aged children and was consistent with infection intensity versus uninfected children.

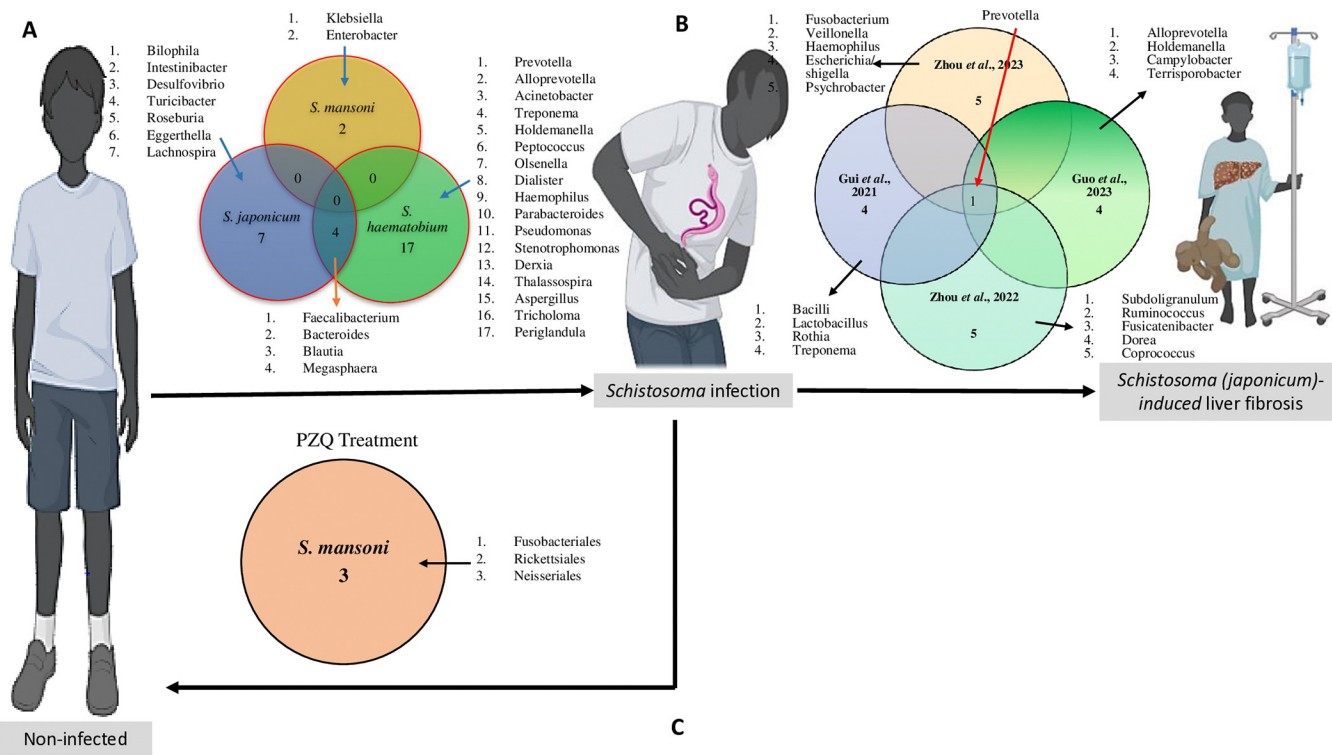

**Fig 2. Flow diagram describing all current human-based studies on the host gut microbiome and schistosomiasis. A.** Intersects of microbes revealed in studies comparing the differential gut microbiomes between non-infected and infected individuals. **B.** Intersects of microbes revealed in studies unveiling the differential gut microbiomes of schistosomiasis patients with liver fibrosis. **C.** Intersects of microbes revealed in studies unveiling the differential gut microbiomes of schistosomiasis patients following praziquantel administration. Cartoon illustrations were retrieved from Biorender.com and used in the present figure.

## Gut microbiome signature in *S. japonicum*-infected patients

The relationship between *S. japonicum* infection and gut microbiome was explored in 3 out of 13 articles: Gordon *et al.* [27], Jiang *et al.* [28] and Lin *et al.* [30]. The two studies reported that Proteobacteria levels increased while Firmicutes levels decreased in response to *S. japonicum* infection. Additionally, the phylum TM7 showed an increased relative abundance in patients with acute *S. japonicum* infection. At the genus level, Jiang *et al.* [28] observed a significant low relative abundance of 5 genera namely: *Comamonas*, *Psychrobacter*, *Clostridium*, *Veillonella*, and *Butyricimonas* with a significant high relative abundance of 2 genera; *Methylophilus* and *Turicibacter*, in this study population. Importantly, gut bacterial genera, such as *Bacteroides*, *Blautia*, *Bilophila*, *Enterococcus*, *Intestinibacter* and *Desulfovibrio* were considered as a predictor for distinguishing *S. japonicum*-infected and uninfected humans based on a machine-learning strategy [30]. See **Table 2**.

## Gut microbiome signature in *S. mansoni*-infected patients

Till date, only Schneeberger *et al.* [37] explored the association between gut microbiome disturbance and *S. mansoni* infection. This study revealed that bacteria from the genera *Klebsiella* and *Enterobacter* were significantly more abundant in *S. mansoni*-infected pre-school-aged and school-aged children. Over-abundance of members from the families *Cerasicoccaceae*, *Anaeroplasmataceae*, *Campylobacteraceae* and *Peptococcaceae*, and of the genus *Fructobacillus* seem to be significantly linked to the faecal microbiome of schistosome-negative children (**Table 2**).

**Table 2. Main findings reported in the included studies.**

| Studies | Studied value for Schistosomiasis | Genera that changed in the microbiome of *Schistosoma*-infected subjects | |
|---|---|---|---|
| | | **Enriched** | **Depleted** |
| [26] | Infection | *Megasphaera, Desulfovibrior, Peptococcus, Olsenella, Haemophilus, Dialister, Alloprevotella, Prevotella, Acinetobactern* and uncultured *Coriobacteriaceae* | *Peptostreptococcaceae incertae sedis, Ruminococcaceae incertae sedis,* uncultured *Erysipelotrichaceae,* uncultured *Mollicutes RF9, Clostridium sensu stricto 6, Parabacteroides* and *Subdoligranulum* |
| [27] | Infection | *Faecalibacterium* | / |
| [30] | Infection | *Blautia, Intestinibacter, Eggerthella, Enterococcus, Ruminiclostridium,* uncultured bacterium *Ruminococcaceae* | *Pantoea, Sutterella, Parasutterella* |
| [31] | Infection | *Pseudomonas, Stenotrophomonas, Derxia, Thalassospira, Aspergillus, Tricholoma* and *Periglandula* | *Azospirillum* |
| [32] | Infection | *Aspergillus, Candida, Enterocytozoon, Protomyces, Entomophthora Taphrina, Saccharomyces.* | |
| [37] | Infection and Treatment | *Klebsiella* and *Enterobacter arachidis, Fusobacterium* | *Fructobacillus* |
| [33] | Pathology | *Faecalis* bacterium and *Bacteroides* | *Prevotella* 9 |
| [36] | Infection and treatment | *Acinetobacter, Treponema, Bacteroides, Alloprevotella* and *Prevotella*; | *Pediococcus, Chloroplast, Lactobacillus* and *Weissella* |
| [38] | Pathology | *Lactobacillus, Streptococcus, Prevotella, Enterococcus, Staphylococcus, Rothia, Treponema, Actinomyces.* | *Megamonas, Lachnospira,* |
| [28] | Infection | *Comamonas, Psychrobacter, Clostridium, Veillonella,* and *Butyricimonas* | *Methylophilus* and *Turicibacter* |
| [29] | Infection and treatment | *Prevotella* | |
| [34] | Pathology | *Prevotella 9, Megamonas, Fusobacterium, Proteobacteria, Veillonella, Streptococcus, Haemophilis, Psychrobacter, Escherichia, Shigella* | *Faecalibacterium, Ruminococcaceae UCG 014* |
| [35] | Pathology | *Prevotella 7, Campylobacter, Alloprevotella, Collinsella, Terrisporobacter, Holdemanella, Streptococcus, Eubacterium ventriosum* group | *Megamonas, Lysobacter, Citrobacter, Dubosiella* |

## Prognostic potential of the gut microbiome for schistosomiasis

The modification of the gut microbiome associated with schistosomiasis-related pathologies were assessed by Zhou *et al.* [33], Zhou *et al.* [34], Guo *et al.* [35] and Gui *et al.* [38] (**Table 2**). All these studied focused on schistosomiasis *japonicum*-induced liver fibrosis and reported difference in gut bacterial community structures between chronic schistosomiasis patients and controls groups with a slight tendency toward a lower bacterial diversity in *S. japonicum* infection-induced liver cirrhosis group [38]. Bacterial relative abundance was evaluated at different taxonomic levels. At the phylum level, analysis revealed that Bacteroidetes, Firmicutes, and Proteobacteria were the three most prevalent phyla in both patients and controls groups [33,38]. However, compared with healthy individuals, patients with chronic infection had a lower abundance of Firmicutes and a higher abundance of Proteobacteria [34]. Patients with cirrhosis show a significant increase in the Bacilli class and Lactobacillales order in their gut microbiota compared to healthy individuals [38]. At the family level, the dominant flora in patients with chronic *S. japonicum* infection was Bacteroidaceae; while, the Prevotellaceae only existed in patients with chronic *S. japonicum* infection [32]. At the genus level, three major taxa, *Bacteroides*, *Blautia*, and *Enterococcus*, were found to be associated with the level of liver injuries induced by *S. japonicum* and could be used for the prediction of schistosomiasis in humans [34]. Five bacterial genus have potential to act as non-invasive biomarkers for differentiation of different stages of *S. japonicum* pathology including *Prevotella*, *Subdoligranulum*, *Ruminococcus*, *Megamonas* and *Fusicatenibacter* [33].

Overall, the above data revealed the genus *Prevotella* as a non-invasive biomarker of schistosomiasis *japonicum*-induced liver pathology (**Fig 2B**).

## Association between gut microbiome and praziquantel treatment during schistosomiasis

The effect of the PZQ treatment on the structure and composition of the gut microbiome was the focus of 3 studies (**Fig 2C**): Kay *et al.* [29], Ajibola *et al.* [36] and Schneeberger *et al.* [37]. Two of these reported that the treatment with PZQ remove the parasites but has little to no effect on the gut microbiome of schistosome-infected subjects compared to changes induced by the infection [29,36]. However, the third study [37] showed that an increased abundance of bacteria from the orders Neisseriales, Rickettsiales and Fusobacteriales is associated with higher efficacy of PZQ in treating *S. mansoni* infections. The findings suggest that *Fusobacterium* spp. abundance is closely linked to *S. mansoni* infection and that the initial presence of *Fusobacterium* spp. may have an impact on the effectiveness of schistosomiasis treatment. See **Table 2**.

## Functional changes associated with Gut Microbiome alterations in schistosomiasis patients

Two KEGG pathways significantly enriched in schistosomiasis-positive: atrazine degradation and Arginine and Proline metabolism, as well as five KEGG orthologs: three subunits of bacterial urease (ureA, ureB, and ureC), prdB (D-proline reductase) and prdF (proline racemase) were identified by Ajibola *et al.* [26]. Osakunor *et al.* [31] found 262 AMR genes, belonging to 12 functional drug class levels; and AMR genes belonging to tetracycline was the most common, followed by beta-lactam, macrolide, sulfonamide and nitroimidazole. Of these, the most abundant genes were cfxA6, followed by tet(Q), tet(W), sul2, erm(F) and nimE. PERMANOVA analysis did not show any significant association of AMR genes with age *S. haematobium* infection, previous praziquantel treatment and antibiotic use on AMR genes [31].

Serum globulin levels, previously identified as a reliable biomarker of inflammation and the immune status, did not change in the context of *S. japonicum* infection-induced liver cirrhosis. This finding suggests that *S. japonicum* infection-induced liver cirrhosis may have a limited impact on systemic immune functions [38].

## Discussion

During the last decade, human gut microbiome studies have garnered increasing attention in order to determine their diagnostic, prognostic and therapeutic potential. In recent decade, several studies were undertaken in order to characterise the gut microbiome signature of schistosomiasis-infected patients with or without associated liver pathology [26–38]. This systematic review aimed to assess the current evidence on the interaction between the gut microbiome and the human schistosomiasis.

Generally, the human gut microbiota is mainly composed of 6 phyla of bacteria including Firmicutes, Bacteroidetes, Actinobacteria, Proteobacteria, Fusobacteria and Verrucomicrobia, which account for over >95% of the gut microbiota [39]. Among the fungi, the most studied are *Candida*, *Saccharomyces*, *Malassezia*, and *Cladosporium* [40]. In addition to bacteria and fungi, the human gut microbiota also contains viruses, phages, and archaea, mainly *Methanobrevibacter smithii* [41]. Data collected from studies included in this systematic review reveal infection with schistosomes lead to a change in diversity of the gut microbiome of schistosomiasis-infected patients, consistent with known influence factors such as age, gender, ethnicity,

genetics, and pathology [42]. It was observed that both *S. haematobium* [27] and *S. japonicum* [36–38] infection significantly reduced the alpha and beta-diversity of infected patients compared to healthy individuals. The reduction of the diversity of the intestinal microbiota leads to intestinal dysbiosis and a decrease in both metabolic and immune function, which can lead to more serious complications [43].

Firmicutes and Proteobacteria are the most affected gut microbiota phyla following schistosomiasis-infection and associated pathology. Studies reported decreased abundance of Firmicutes and increased abundance of Proteobacteria. Several studies have linked these gut microbiome changes with the inflammatory environment [44–47]. Indeed, members of phylum Firmicutes are known for their ability in metabolizing complex carbohydrates, which can promote the induction of colonic regulatory T (Treg) cells. These Treg cells are crucial for suppressing inflammatory and allergic responses [48]. A reduction in Firmicutes abundance could thus contribute to a pro-inflammatory environment in the gut. Moreover, elevated levels of Proteobacteria-derived lipopolysaccharides (LPS) can trigger immune responses in the gut, leading to inflammation [49]. This inflammation environment in the gut can exacerbate the pathology associated with the schistosomiasis infection. Jiang *et al.* [32] reported an increase in relative abundance of TM7 phylum in acute *S. japonicum* infected patients, which is commonly associated with inflammatory diseases, and the TM7 phylotype is recognize for its potential immunosuppressive capacity through the inhibition of the production of tumour necrosis factor alpha (TNF-α) [50].

Many bacteria genera are considered as biomarkers to distinguish between infected and uninfected humans, mainly *Bacteroides*, *Faecalibacterium*, *Blautia* and *Megasphaera*. Members of the genus Bacteroides such as *Bacteroides vulgatus* and *Bacteroides fragilis* have been reported to be the two main isolates from patients suffering from Crohn's disease, while the latter has been associated with intra-abdominal abscesses, appendicitis, and inflammatory bowel disease [49]. Data suggest that Enterotoxigenic *Bacteroides fragilis* secretes a toxin termed *B. fragilis* toxin (BFT) which induces IL-8 secretion from intestinal epithelial cell lines including HT29, T84, and Caco-2. IL-8 is a major chemotactic and activating peptide for neutrophils, and increased IL-8 expression is found in both acute and chronic inflammation [50]. Many studies have shown that *Faecalibacterium* abundance is reduced in different intestinal disorders and may be useful potential biomarker to assist in the diagnostic of gut diseases such as ulcerative colitis and Crohn's disease [51]. Indeed, anti-inflammatory properties have been attributed to *F. prausnitzii*, through its capability to induce a tolerogenic cytokine profile with very low secretion of pro-inflammatory cytokines like IL-12 and IFN-γ, and an elevated secretion of the anti-inflammatory cytokine IL-10 [52]. Benítez-Páez *et al.* [53] reported that decreased abundance of *Blautia* species, especially *Blautia luti* and *B. wexlerae*, in the gut microbiota of obese children, are correlated with increase in pro-inflammatory cytokines and chemokines (gamma interferon [IFN-γ], tumour necrosis factor alpha [TNF-α], and monocyte chemoattractant protein 1 [MCP-1]).

The gut microbiome has been extensively investigated for the understanding of the physiopathology of many diseases. Changes in the composition and function of gut microbiota are correlated with the development and progression of multiple kinds of liver diseases [54–57]. To date, the role of the gut microbiome in the schistosomiasis-induced pathology is understudied; available data are limited to liver injuries induced by *S. japonicum* [35–38]. In this review, studies reported mainly *Prevotella*, genus as potential non-invasive biomarkers for differentiation of different stages of *S. japonicum* pathology. This genus is a common feature of the human gut microbiome and have been repeatedly implicated in health and disease. The genus *Prevotella* is a common component of the human gut microbiome and frequently linked to various health conditions. Within this genus, there are two main lineages: one includes *P.*

*copri*, while the other comprises *P. stercorea*. The latter lineage produces sialidases, which are involved in immune cell interactions, pathogen binding, and gut inflammation [58,59]. Furthermore, the exposure of the liver to whole bacteria and/or their products such as LPS, peptidoglycan, viral or bacterial DNA, or fungal beta-glucan, due to compositional change of gut microbiome, leads to the induction of proinflammatory changes in the liver. These components, collectively labelled microbe-associated molecular patterns (MAMPs), are then recognised by liver innate immune cells (Kupffer cells, dendritic cells, NK and NKT cells, and hepatic stellate cells). Antigens derived from the microbiota induce inflammation by binding to pattern recognition receptors (PRRs) on liver macrophages, including Kupffer cells and stellate cells. Signalling via PRRs, mostly TLRs, leads to increased production of inflammatory (TNF, IL-1, IL-6) and fibro-genic cytokines/chemokines (TGF, MCP-1) as well as oxidative and endoplasmic reticulum (ER) stress. Microbial antigens can also induce type I interferon responses in the liver, leading to proliferation and activation of CD8+ cytotoxic T cells. Other immunological mechanisms, such as various effects of short-chain fatty acids on adaptive immune responses, are discussed separately. All these immunological mechanisms may contribute to the development of inflammation-mediated liver injury, which may progress to fibrosis, cirrhosis [60].

Few studies have investigated the effect of PZQ treatment on the gut microbiome, and some of them fail to find any relationship between schistosomiasis treatment and changes in the gut microbiome composition. In the only study that found that PZQ does in fact have a slight short-term effect on the gut microbe, authors reported that members of the class Fusobacteriales were significantly more abundant in the successfully treated group at baseline [34]. This genus is particularly interesting as it includes species known to have inflammatory properties and species that have been linked to chronic disease, such *as Fusobacterium nucleatum*, that could be closely associated with the worm-triggered inflammatory response. The high abundance of *Fusobacterium* spp. before treatment and their high decrease over the first 24 hours post-treatment is correlated with a better outcome of praziquantel treatment. This could indicate that the abundance of the genus *Fusobacterium* and the *S. mansoni* infection are dynamically linked and that preliminary presence of *Fusobacterium* correlated positively with the treatment efficacy following PZQ administration but were not simply altered as a general result of PZQ administration [34].

## Conclusion

At the end of this systematic review, the existing data on the relationship between the gut microbiome and schistosomiasis in humans clearly showed that human schistosomiasis altered significantly the structure and composition of the human gut microbiome leading to dysbiosis state. The observation of shared gut microbial features across *Schistosoma* spp. infestations suggest the potential of gut microbiome to provide consensus non-invasive biomarker(s) for detecting schistosomiasis, related pathological conditions and predicting PZQ efficacy.

## Outlook

Our present review, considering the depth of the available data, although confirming a clear diagnostic, prognostic and therapeutic biomarking potential of the human gut microbiome during schistosomiasis, cannot yet present some conclusive profiles of gut microbiome inherent to this biomarking. Indeed, this would require our analytical models aimed at identifying such robust biomarkers to be fed further data

from large-scale, longitudinal, diet-controlled and poly- versus mono-infected studies of more gut microbial profiles in schistosomiasis-infested individuals. Such research could pave

the way for new management strategies rooted into the identification of unique and pathogno-monic gut microbial features of schistosomiasis at both the infection as well as the pathological levels.

Notably somehow in terms of limitation of the currently available data sets, the view offered by most studies, reported in this review, have made use of 16S rRNA gene sequencing on highly heterogenous and superficially categorized individuals that targets at a limited depth only bacteria and archaea. Future comprehensive studies would benefit from the use of more in-depth, comprehensive and untargeted sequencing approaches such as whole genome shot-gun metagenomics to include a more refined level of appraisal of gut microbe profiles, includ-ing non-bacterial microbes, in achieving the biomarking of schistosomiasis infections and associated morbidities. This will necessitate the use of robustly clinically categorized hosts from endemic areas i.e. with strictly defined clusters of infection statuses and burden and con-trolled for sources of bias such as age, gender, exposure experience, nutrition and, where feasi-ble, genetic make-up.

## Supporting information

**S1 Table. PRISMA 2020 Checklist.**
(DOCX)

**S2 Table. Search strategy through Medline, Embase, Global Health, Web of Science, and Global Index Medicus databases.**
(DOCX)

**S3 Table. Quality assessment of the studies.**
(XLSX)

## Acknowledgments

The authors are grateful to Merck KGaA Global Health Institute i.e. Drs Thomas Spangenberg and Claudia Demarta-Gatsi for critically revising the present review and assisting in defining the manuscript plan.

## Author Contributions

**Conceptualization:** Martin Gael Oyono, Justin Komguep Nono.

**Formal analysis:** Sebastien Kenmoe, Jean Thierry Ebogo Belobo, Mireille Kameni, Thabo Mpotje.

**Funding acquisition:** Justin Komguep Nono.

**Investigation:** Jean Thierry Ebogo Belobo, Leonel Javeres Mbah Ntepe, Leonel Meyo Kamguia.

**Methodology:** Martin Gael Oyono, Sebastien Kenmoe, Jean Thierry Ebogo Belobo.

**Supervision:** Justin Komguep Nono.

**Writing – original draft:** Martin Gael Oyono, Sebastien Kenmoe, Jean Thierry Ebogo Belobo, Leonel Javeres Mbah Ntepe, Mireille Kameni, Leonel Meyo Kamguia.

**Writing – review & editing:** Martin Gael Oyono, Thabo Mpotje, Justin Komguep Nono.

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
