## [Decision Letter · Decision Letter 0]

2 Oct 2024

Dear Dr Nono,

Thank you very much for submitting your manuscript "Diagnostic, Prognostic, and Therapeutic Potentials of Gut Microbiome Profiling in Human Schistosomiasis: A Comprehensive Systematic Review" for consideration at PLOS Neglected Tropical Diseases. As with all papers reviewed by the journal, your manuscript was reviewed by members of the editorial board and by several independent reviewers. In light of the reviews (below this email), we would like to invite the resubmission of a significantly-revised version that takes into account the reviewers' comments.

We cannot make any decision about publication until we have seen the revised manuscript and your response to the reviewers' comments. Your revised manuscript is also likely to be sent to reviewers for further evaluation.

Sincerely,

Timir Tripathi, Ph.D.

Academic Editor

Jong-Yil Chai

Section Editor

Reviewer's Responses to Questions

**Key Review Criteria Required for Acceptance?**

**Methods**

-Are the objectives of the study clearly articulated with a clear testable hypothesis stated?

-Is the study design appropriate to address the stated objectives?

-Is the population clearly described and appropriate for the hypothesis being tested?

-Is the sample size sufficient to ensure adequate power to address the hypothesis being tested?

-Were correct statistical analysis used to support conclusions?

-Are there concerns about ethical or regulatory requirements being met?

Reviewer #1: The aim of the review was well stated line 134

The design of the systematic review was well outlined with a standard PRISMA based checklist. The required tools for literature search, database coverage, data extraction and selection for inclusion in the review were used.

The review sought to identify microbiome enrichment in different Schistosomiasis species infections and hence the appropriate statistical tools were used to compare the retrieved data inorder to address the objectives

A statement on ethical considerations is availed showing that it is not required for a systematic review.

Reviewer #2: (No Response)

Reviewer #3: There are several issues with the Methods section:

1. In the "Assessment of Risk of Bias," the authors should specify which JBI checklists were used. Since they included various types of study designs, these cannot be assessed with a single quality assessment checklist. Additionally, because they incorporated websites into their search, they should clarify the details of the findings derived from these websites, as well as the checklist used to assess their quality.

2. The PRISMA flowchart (Figure 1) requires further clarification regarding "title and abstract screening," "full text screening," and "included studies." It is unclear why the flowchart mentions both "studies included in review (n=13)" and "reports of included studies (n=13)" in the final section.

**Results**

-Does the analysis presented match the analysis plan?

-Are the results clearly and completely presented?

-Are the figures (Tables, Images) of sufficient quality for clarity?

Reviewer #1: The analysis and results were well presented with figure shematic of the comparative analysis of the microbiomes based on schistosomiasis species infection. The tables in the main text and supplementary materials do have the source information that was necessary for the analysis.

Reviewer #2: Most sections of the results section must be rewritten

Reviewer #3: There are several issues with the Results section:

1. There are missing references throughout the Results section. The authors should cite the references for the included studies when reporting their characteristics. For example, in line 235, they should reference the "cross-sectional studies" when mentioning their number.

2. In line 242, the sentence "In SSA, the most affected part worldwide" is not a result of their study. The authors should either provide a reference for this statement or delete it, as the Results section should only include findings from the included studies.

3. There is a significant amount of heterogeneous information in the tables. For example, in Table 1, it is expected to summarize the "study population" coherently (e.g., how many patients were examined). Additionally, there is missing information regarding the demographic characteristics of the studied population (e.g., gender, age, etc.) that should be presented in separate columns in Table 1. Table 2 also contains excessive uncategorized information. In systematic reviews, it is important to present principal information clearly and concisely in tables.

4. Figure 2 has very low quality and should be enhanced for clarity.

5. There are several misclassifications of bacterial taxa. For example, in line 358, "Neisseriales," "Rickettsiales," and "Fusobacteriales" are not classes but orders. The authors should review the manuscript carefully and address all necessary corrections.

6. Correct notation should be used for bacterial taxa. For instance, in line 392, "Firmicutes" is a kingdom and should not be italicized. There are many other items to address. The authors can refer to "https://wwwnc.cdc.gov/eid/page/scientific-nomenclature" for guidance on reporting bacterial names. Note that kingdom, phylum, class, order, and suborder should begin with a capital letter but should not be italicized.

**Conclusions**

-Are the conclusions supported by the data presented?

-Are the limitations of analysis clearly described?

-Do the authors discuss how these data can be helpful to advance our understanding of the topic under study?

-Is public health relevance addressed?

Reviewer #1: The conclusions are supported by the systematic review and limitations are clearly stated.

In the discussion, the systematic review profiling the microbial taxa in Schistosoma species infections, was well supported by citations of works with similar findings and possible implication of the enriched microbiome taxa

Reviewer #2: The conclusion cannot be matched with the objective of the study as well as with the title of the manuscript

Reviewer #3: There are several issues with the Discussion and Conclusions sections:

1. The Conclusion section is too lengthy. I suggest summarizing this part to highlight the key findings of the study.

2. In lines 387-389: "In recent decades, several studies were undertaken to characterize the gut microbiome signature of schistosomiasis-infected patients with or without associated liver pathology." Please provide references for this statement and cite the mentioned studies.

**Editorial and Data Presentation Modifications?**

Reviewer #1: Line 41: replace "i.e" with "that is"

Line 69: This neglected tropical disease affects atleast 251.4 million people

Line 230: Replace 'On' with 'Of' in the sentence "Of the 24 articles...."

Line 390: Also add that "in order to determine their diagnostic, prognotsic and therapuetic potential."

Reviewer #2: I can submit the PDF of this manuscript in which the minor and majors points have been underlined

Reviewer #3: The manuscript would benefit from language revisions to enhance fluency. For example, in line 385, the phrase 'The studying of human gut microbiome has received increasing attention because...' is awkward. It can be modified to 'Human gut microbiome studies have garnered increasing attention in recent years.' Please review and revise the manuscript accordingly.

**Summary and General Comments**

Reviewer #1: The systematic review was well written however a few points that need to be included in the conclusion;

- A need to discuss the data that was missing from these studies which would influence the outcome. For instance, the general assumption is that the participants/patients had monoinfection and hence no report on the cofounder effects of comorbidities and confections that could potentially influence the outcome.

- The diet of the participants. The diet plays a key role in the structure of the gut microbiome. The food composition was worth reporting since this is not a constant for all the communities from the different geographical locations

Reviewer #2: This manuscript PNTD-D-24-01193 of Oyono et al entitled "Diagnostic, Prognostic, and Therapeutic Potentials of Gut Microbiome Profiling in Human Schistosomiasis: A Comprehensive Systematic Review" treat an interesting topic by reviewing published article dealing with microbiome modifications during schistosome infections.

Although the title is very interesting, the manuscript content does not reflect the title. In the manuscript, the authors tried to identify the diversity of microorganims between schistosomiasis infected and uninfected and also between patients presenting different schistosomiasis-related pathologies. These authors also tried discuss the impacts of the presence of some specific microorganisms with immnunological and clinical outcome of schistosome infections in patients. The pathological implications of the presence of these microorganisms were also discussed. In terms of diagnostic, prognostic and therapeutic potentials of these microorganisms, no data can be inferred. I was expected to see a list of potential biomarkers that could open a framework for investigations aiming to develop new diagnostic or prognostic or therapeutic tools. In their study, the authors reported diversity of gut microorganisms with some of them that are probably abundant when the infected patients present some pathological manifestations. However, as these microorganisms could be found in patients suffering from other diseases, could their detection be considered as potential indication of schistosome infections and the pathology associated to these infections? This article could be focused on : A Comprehensive Systematic Review on gut Microbiome Profiling in Human Schistosomiasis for the understanding of schistosome infections and associated pathologies. 

The manuscript is not well written and the language requires deep revision

Overall, this manuscript contains a lot of shortcomings and in the present form, it cannot be accepted for publication in Plos NTD

Reviewer #3: (No Response)

PLOS authors have the option to publish the peer review history of their article (what does this mean?). If published, this will include your full peer review and any attached files.

Reviewer #1: No

Reviewer #2: Yes: Gustave SIMO

Reviewer #3: Yes: Parnian Jamshidi
---

## [Decision Letter · Decision Letter 1]

9 Jan 2025

PNTD-D-24-01193R1Diagnostic, Prognostic, and Therapeutic Potentials of Gut Microbiome Profiling in Human Schistosomiasis: A Comprehensive Systematic ReviewPLOS Neglected Tropical Diseases

Dear Dr. Nono, Thank you for submitting your manuscript to PLOS Neglected Tropical Diseases. After careful consideration, we feel that it has merit but does not fully meet PLOS Neglected Tropical Diseases's publication criteria as it currently stands. Therefore, we invite you to submit a revised version of the manuscript that addresses the points raised during the review process.

Please submit your revised manuscript within 30 days Feb 08 2025 11:59PM. If you will need more time than this to complete your revisions, please reply to this message or contact the journal office at plosntds@plos.org. Please include the following items when submitting your revised manuscript: *
A rebuttal letter that responds to each point raised by the editor and reviewer(s). You should upload this letter as a separate file labeled 'Response to Reviewers'. This file does not need to include responses to any formatting updates and technical items listed in the 'Journal Requirements' section below. *
A marked-up copy of your manuscript that highlights changes made to the original version. You should upload this as a separate file labeled 'Revised Manuscript with Track Changes'. *
An unmarked version of your revised paper without tracked changes. You should upload this as a separate file labeled 'Manuscript'. If you would like to make changes to your financial disclosure, competing interests statement, or data availability statement, please make these updates within the submission form at the time of resubmission. Guidelines for resubmitting your figure files are available below the reviewer comments at the end of this letter. We look forward to receiving your revised manuscript. Kind regards, Timir Tripathi, Ph.D.Academic EditorPLOS Neglected Tropical Diseases Jong-Yil ChaiSection EditorPLOS Neglected Tropical Diseases

Shaden Kamhawi

co-Editor-in-Chief

Paul Brindley

co-Editor-in-Chief

**Additional Editor Comments:** One of the original reviewer has suggested minor revisions in your manuscript.

**Journal Requirements:**

Please add in-text citation for Table 2 within your manuscript.

**Reviewers' comments:** Reviewer's Responses to Questions

**Key Review Criteria Required for Acceptance?**

**Methods**

-Are the objectives of the study clearly articulated with a clear testable hypothesis stated?

-Is the study design appropriate to address the stated objectives?

-Is the population clearly described and appropriate for the hypothesis being tested?

-Is the sample size sufficient to ensure adequate power to address the hypothesis being tested?

-Were correct statistical analysis used to support conclusions?

-Are there concerns about ethical or regulatory requirements being met?

Reviewer #2: This section is well described

Reviewer #3: (No Response)

**Results**

-Does the analysis presented match the analysis plan?

-Are the results clearly and completely presented?

-Are the figures (Tables, Images) of sufficient quality for clarity?

Reviewer #2: Results have been well presented

Reviewer #3: (No Response)

**Conclusions**

-Are the conclusions supported by the data presented?

-Are the limitations of analysis clearly described?

-Do the authors discuss how these data can be helpful to advance our understanding of the topic under study?

-Is public health relevance addressed?

Reviewer #2: The conclusion is acceptable now

Reviewer #3: (No Response)

**Editorial and Data Presentation Modifications?**

Reviewer #2: - Lines 115- 117: References must be added

- The size of figure must be increased

- Line 238-242: These sentences must be reviewed

- Line 244: One "burden" must be deleted

- Lines 345-347: This sentence must be reviewed

Reviewer #3: (No Response)

**Summary and General Comments**

Reviewer #2: The manuscript has been deeply improved

Reviewer #3: The authors have effectively addressed all the comments and revisions.

PLOS authors have the option to publish the peer review history of their article (what does this mean?). If published, this will include your full peer review and any attached files.

Reviewer #2: **Yes: **Gustave Simo

Reviewer #3: **Yes: **Parnian Jamshidi

**Figure resubmission:** While revising your submission, please upload your figure files to the Preflight Analysis and Conversion Engine (PACE) digital diagnostic tool, https://pacev2.apexcovantage.com/. PACE helps ensure that figures meet PLOS requirements. To use PACE, you must first register as a user. Registration is free. Then, login and navigate to the UPLOAD tab, where you will find detailed instructions on how to use the tool. If you encounter any issues or have any questions when using PACE, please email PLOS at figures@plos.org. Please note that Supporting Information files do not need this step. If there are other versions of figure files still present in your submission file inventory at resubmission, please replace them with the PACE-processed versions.

**Reproducibility:** To enhance the reproducibility of your results, we recommend that authors of applicable studies deposit laboratory protocols in protocols.io, where a protocol can be assigned its own identifier (DOI) such that it can be cited independently in the future. Additionally, PLOS ONE offers an option to publish peer-reviewed clinical study protocols. Read more information on sharing protocols at https://plos.org/protocols?utm_medium=editorial-email&utm_source=authorletters&utm_campaign=protocols

---

## [Editor Report · Decision Letter 2]

16 Jan 2025

Dear Dr Nono,

We are pleased to inform you that your manuscript 'Diagnostic, Prognostic, and Therapeutic Potentials of Gut Microbiome Profiling in Human Schistosomiasis: A Comprehensive Systematic Review' has been provisionally accepted for publication in PLOS Neglected Tropical Diseases.

Best regards,

Timir Tripathi, Ph.D.

Academic Editor

Jong-Yil Chai

Section Editor

Shaden Kamhawi

co-Editor-in-Chief

Paul Brindley

co-Editor-in-Chief

Thank you for the revision of your manuscript. It is now accepted for publication.

---

## [Editor Report · Acceptance letter]

25 Jan 2025

Dear Dr Nono,

We are delighted to inform you that your manuscript, "Diagnostic, Prognostic, and Therapeutic Potentials of Gut Microbiome Profiling in Human Schistosomiasis: A Comprehensive Systematic Review," has been formally accepted for publication in PLOS Neglected Tropical Diseases.

Best regards,

Shaden Kamhawi

co-Editor-in-Chief

Paul Brindley

co-Editor-in-Chief
